# Adult Attachment and Emotion Regulation Flexibility in Romantic Relationships

**DOI:** 10.3390/bs14090758

**Published:** 2024-08-27

**Authors:** Farnaz Mosannenzadeh, Maartje Luijten, Dominique F. MacIejewski, Grace V. Wiewel, Johan C. Karremans

**Affiliations:** 1Behavioural Science Institute, Radboud University, 6525 XZ Nijmegen, The Netherlands; farnaz.mosannenzadeh@ru.nl (F.M.); maartje.luijten2@ru.nl (M.L.); grace.wiewel@ru.nl (G.V.W.); 2Tilburg School of Social and Behavioral Sciences, Tilburg University, 5037 AB Tilburg, The Netherlands; d.f.maciejewski@tilburguniversity.edu

**Keywords:** adult attachment, emotion regulation, flexibility, romantic relationships, ESM

## Abstract

Adults with attachment insecurity often struggle in romantic relationships due to difficulties in emotion regulation (ER). One potentially influential yet understudied factor is the inflexible over-reliance on either intrapersonal (self-directed, e.g., suppression) or interpersonal (involving others, e.g., sharing) ER. This study investigates the association between attachment insecurity and flexibility in using interpersonal versus intrapersonal ER in response to daily stressors in romantic relationships. We hypothesized that higher attachment avoidance and anxiety are associated with (H1) higher reliance on either intrapersonal or interpersonal ER over the other, respectively; (H2) less variable use of interpersonal compared to intrapersonal ER over time; and (H3) less flexible use of interpersonal compared to intrapersonal ER depending on the availability of a romantic partner. Study 1 (*N* = 174; 133 females, *M*_age_ = 23.79, *SD*_age_ = 7.63) used an online cross-sectional survey to measure average inter/intrapersonal ER, addressing H1. Study 2 (*N* = 124; 104 females, *M*_age_ = 22.45, *SD*_age_ = 6.39), combined a baseline survey with experience sampling (7 days, 8 notifications/day), addressing H1, H2, and H3. Results showed that higher attachment avoidance was associated with lower interpersonal compared to intrapersonal ER. Higher attachment anxiety was associated with less variable use of interpersonal compared to intrapersonal ER and less flexible use of interpersonal ER depending on partner availability. These findings suggest distinct associations between attachment orientations and ER flexibility, explaining ER difficulties in individuals with high attachment insecurity.

## 1. Introduction

Most people seek a fulfilling romantic relationship, which is known to enhance health and longevity [1]. However, adults with higher attachment insecurity have difficulties in the formation and maintenance of satisfying romantic relationships partially due to difficulties in regulating their emotions [2,3,4,5]. Understanding how attachment insecurity is associated with difficulties in the regulation of emotions is a key step to enhancing individual well-being and nurturing more satisfying relationships for people with higher attachment insecurity.

Emotion regulation (ER) refers to all processes through which individuals change their emotions; for example, reducing negative emotions [6]. Strategies for ER range from distracting oneself to seeking support from others [6,7]. While extensive research has examined the general effectiveness of various ER strategies [8], the more recent literature has recognized that successful ER is not necessarily about using *particular* ER strategies, but depends on the ability to flexibly use *a variety of different* ER strategies that suit specific personal and situational demands [9,10]. This ability is termed emotion regulation flexibility (ER flexibility) [9,10,11].

Higher attachment insecurity has been linked to lower cognitive and psychological flexibility [12,13,14], raising the question whether it may also be related to limited ER flexibility. As we will explain in more detail, individuals with higher attachment insecurity may struggle with seeking support from others for emotion regulation, resulting in either excessive self-reliance or dependency on others [5,15,16]. This over- or under-dependence on support from others may suggest a limited ER flexibility. This is important because limited ER flexibility may at least to some extent explain and thus provide a better understanding of ER difficulties of adults with higher attachment insecurity. However, previous research has not directly investigated the association between adult attachment and ER flexibility. The present research aims to investigate if and to what extent attachment insecurity is associated with ER flexibility in the context of romantic relationships.

### 1.1. Adult Attachment and Emotion Regulation in Romantic Relationships

According to attachment theory [17,18,19], attachment is an innate behavioral system designed to ensure a person’s safety through proximity to a caregiver, known as the attachment figure [3]. Individuals with a more secure attachment hold positive views of themselves as worthy of love and view close others as being reliable in times of need [3,20]. In contrast, insecure attachment is marked by self-doubt and skepticism towards other’s reliability in times of need [3]. In adulthood, attachment-related dynamics extend into romantic relationships, with romantic partners serving as attachment figures [21,22]. Individual differences in attachment behavior in adulthood are measured by two continuous dimensions (called attachment orientations): attachment anxiety and attachment avoidance [23]. In the context of romantic relationships, higher attachment anxiety is characterized by an exaggerated fear of being abandoned by romantic partners. *Higher attachment avoidance* is characterized by a high reluctance to shape interdependence with romantic partners [3].

Adult attachment orientations play a pivotal role in how individuals regulate their emotions in the context of romantic relationships. In times of stress, the primary attachment-related ER strategy, associated with a secure attachment, is seeking proximity to a romantic partner [5,15,16,22]. For example, John who is having a stressful conflict at work, reaches out to his romantic partner Mary by phone, seeking her support and proximity to deal with his stress. Notably, this primary strategy is a form of interpersonal ER, which refers to the processes in which “a person’s emotions are regulated by others” [24] (p. 342). Interpersonal ER can be contrasted to intrapersonal ER, which refers to regulation of emotions without the help of others [6]. For example, John suppresses his negative emotions or brings his attention to the positive aspects of his work to feel better [25]. In attachment terms, effective interpersonal ER occurs when the attachment figure is available and responsive to one’s needs and requests for support [5,26]; for example, when Mary picks up the phone and assures John that he can well-handle the conflict, making John feel less stressed.

However, if the attachment figure appears unavailable, for example, if Mary does not respond to John’s phone call, individuals may use secondary emotion regulation strategies. These secondary strategies, called hyperactivating and deactivating strategies, are associated with attachment insecurity [5,15,16] and may indicate an over-reliance on either interpersonal or intrapersonal ER over the other.

Hyperactivation occurs when the attachment figure appears unavailable but proximity seeking is still perceived viable [5]. It involves intensification of attachment related emotions (e.g., stress) and behaviors (e.g., proximity seeking) [5,15]. For instance, John becomes angry when Mary does not pick up the phone and keeps calling her. Hyperactivating strategies are typically used by individuals with higher attachment anxiety [5,27,28]. For example, empirical research has found links between higher attachment anxiety and higher rumination on negative feelings [29], self-blame [30], and catastrophizing [30], all leading to more intense negative emotions. Higher attachment anxiety is also linked to high or excessive reassurance seeking [4,31,32] and more interpersonal ER strategies in general [4]. Notably, the underlying drive for hyperactivating strategies seems to be an exaggeration of self-helplessness [33] in order to elicit the attention, proximity, and support of the romantic partner. This suggests that hyperactivating strategies may reflect difficulties in self- or intrapersonal ER and an insistence on interpersonal ER; in other words, an over-reliance on interpersonal compared to intrapersonal ER, particularly when the romantic partner’s availability is under question.

By contrast, the use of deactivation strategies occurs when proximity seeking is perceived inviable [5]. Deactivation includes suppression of attachment-related emotions and inhibition of proximity-seeking behaviors [5]. For example, John, assuming that Mary is incapable or unwilling to be of help, will not seek contact with Mary but instead withdraws and experientially suppresses stressful feelings. People with higher attachment avoidance more often use deactivating strategies [5,28]. For example, empirical research has shown an association between higher attachment avoidance and higher denial or suppression of emotions [5,34,35], as well as less reassurance seeking [4,31], and lower care-seeking in response to a threatening stimuli [36], all indicating more intrapersonal and less interpersonal ER. Notably, the underlying drive for deactivating strategies seems to be dampening interdependence to protect oneself from rejection-related stress and vulnerability [37]. Deactivating strategies are thus characterized by avoiding reliance on others, and “compulsive self-reliance” or autonomy [5,17,18,19]. This suggests that deactivating strategies may reflect an over-reliance on intrapersonal compared to interpersonal ER.

Considering individuals frequently employ multiple ER strategies concurrently [9], simultaneous investigation of how individuals use interpersonal compared to intrapersonal ER would provide a holistic and realistic perspective on individuals’ ER. It would also capture the relative emphasis an individual places on interpersonal compared to intrapersonal ER across different contexts. However, while previous research has investigated the association between adult attachment and interpersonal or intrapersonal ER [4,30,31,34,38,39], a direct examination of the association between adult attachment and the use of interpersonal compared to intrapersonal ER is lacking. Investigating the association between adult attachment and the *relative* use of interpersonal compared to intrapersonal ER is particularly important as it may indicate that those with higher attachment insecurity may have a limited ER flexibility.

### 1.2. Adult Attachment and Emotion Regulation Flexibility in Romantic Relationships

Recent developments in ER research underscore that successful ER depends on ER flexibility: the extent to which one can use a variety of different ER strategies with respect to specific personal and situational demands across time [9,40,41]. Bonanno and Burton [42] proposed that a flexible ER is one that is sensitive to the contextual demands, uses a wide repertoire of ER strategies and is responsive to feedback. Aldao et al. [9] expanded on the work of Bonanno and Burton [42] and stated that a necessary but not sufficient condition for ER flexibility is ER variability, defined as “the variation in the use of one or more ER strategies across a number of situations” [9] (p. 268). In the previous paragraphs, we explained how adult attachment orientations might be associated with the use of interpersonal compared to intrapersonal ER. In the following paragraphs, we explain how that association may suggest a link between attachment and ER variability and ER flexibility.

#### 1.2.1. Emotion Regulation Variability

People can use ER strategies variably. For example, they can use various ER strategies in response to one specific occasion (e.g., John tries to suppress his stress, but if that does not work, he resorts to asking support from Mary), termed between-strategy variability in the ER flexibility literature [9,11]. Using both interpersonal and intrapersonal ER to a similar extent represents high variability, while a preference for one strategy over the other is an indicator of low variability [43]. Additionally, the preference for ER strategies may vary across different occasions over time (e.g., John asks Mary for support when stressed on one occasion, and on the next occasion John tries coping with stress himself) [9,43]. Variations in the use of an ER strategy across situations over time is termed within-strategy variability in the ER flexibility literature [9,11]. We expect that those with higher attachment insecurity show lower variability in the use of interpersonal compared to intrapersonal ER. Specifically, as we discussed above, we hypothesize that attachment anxiety is linked to a strong preference for and use of interpersonal over intrapersonal ER; and attachment avoidance is linked to a strong preference for and use of intrapersonal over interpersonal ER. Since the ER preferences of individuals with attachment insecurity seem to be compulsive, we also expect these preferences not to vary much across different occasions over time.

#### 1.2.2. Emotion Regulation Flexibility

Emotion regulation flexibility indicates the extent to which the variability in the use of ER strategies covaries with changes in the environment or the situational context [9]. In the context of attachment-related ER in romantic relationships, a crucial contextual factor is the availability of the romantic partner [5,21]. In this sense, a flexible ER would indicate varying interpersonal and intrapersonal ER according to changes in partner availability. More specifically, it would indicate using interpersonal ER (e.g., seeking company and support of the partner) when the romantic partner is available, and being able to resort to intrapersonal ER (i.e., dealing with the emotions by oneself, through for example relaxation, distraction, suppression, and so on), when the romantic partner is absent. By contrast, insisting on either interpersonal or intrapersonal ER irrespective of partner availability would indicate limited ER flexibility. As discussed above, people with higher attachment anxiety use hyperactivating ER strategies, relying on interpersonal over intrapersonal ER, even if their partner is unavailable (e.g., John keeps calling Mary for support even when she apparently is not available, and he does not resort to intrapersonal ER instead). In contrast, people with higher attachment avoidance tend to perceive that their romantic partner is unwilling or incapable of being responsive to their needs, and therefore, even if their partner is available, they may tend to use deactivating strategies, relying on intrapersonal over interpersonal ER (e.g., John tries solving his stress about work using intrapersonal ER, irrespective of whether Mary is available to support him). Thus, variation in the relative use of interpersonal compared to intrapersonal ER for people with higher attachment insecurity should be less dependent on the availability of their partner, indicating a limited ER flexibility.1.3. The Current Research

In the current research, we aim to investigate if and to what extent romantic attachment insecurity is associated with ER flexibility in the use of interpersonal compared to intrapersonal ER when regulating one’s own negative emotions in times of stress. To address this aim, we first investigated the basic question whether individuals with higher romantic attachment insecurity have a tendency to consistently rely on either inter- or intrapersonal ER, relative to the other, as an indicator of ER variability (Research Question 1; RQ1). We used two strategies to examine this research question: Firstly, we investigated the relative use of interpersonal ER compared to intrapersonal ER, with a strong preference for either one of them indicating low variability. Secondly, we investigated the extent to which the relative use of interpersonal compared to intrapersonal ER varies across multiple occasions over time, with less variation indicating low variability.

To measure the relative use of interpersonal compared to intrapersonal ER, in both Studies 1 and 2, in an online cross-sectional survey, after assessing their level of attachment anxiety and avoidance, participants reported their general tendency to use interpersonal ER and intrapersonal ER in response to stressful life events. In addition, in Study 2, we measured the relative use of interpersonal ER compared to intrapersonal ER in multiple occasions per day in a 7-day period, using an experience sampling method (ESM). ESM allows participants to momentarily and repeatedly report on their actual life experiences and behaviors. A meta-analysis comparing global self-reports and daily measures [44] revealed that global self-report measures of a given ER strategy often do not strongly or specifically match the actual use of that strategy in daily life. This finding highlights the advantage of using ESM to capture more accurate and context-specific data on ER. We hypothesized that higher scores on romantic attachment anxiety and avoidance are associated with lower variability, with attachment anxiety being linked to a higher preference for interpersonal over intrapersonal ER (H1.1); attachment avoidance being linked to a higher preference for intrapersonal over interpersonal ER (H1.2); and both attachment anxiety and avoidance being linked to lower variation in the relative use of interpersonal compared to intrapersonal ER across multiple occasions in a given time period (H2.1, H2.2, respectively).

As the second research question (RQ2), we investigated whether the relative use of interpersonal compared to intrapersonal ER varies depending on the availability of the romantic partner, indicating ER flexibility. In general, we hypothesized that the relative use of interpersonal compared to intrapersonal ER would increase when the partner is available compared to when they are absent (H3.1). We then hypothesized that individuals with higher attachment anxiety and/or avoidance will show less ER flexibility, indicated by a weaker association between partner availability and the relative use of interpersonal compared to intrapersonal ER (H3.2; H3.3, respectively).

While our primary focus was on the relative use of interpersonal compared to intrapersonal ER, for each hypothesis, we additionally investigated the unique associations between attachment orientations and these two types of ER separately. This approach allowed us to discern how attachment orientations influence the use of each ER strategy individually, providing more clarity on the origins of ER variability and flexibility.

## 2. Study 1

In this study, we investigated the association between romantic attachment and the general tendency to use interpersonal compared to intrapersonal ER in times of stress using a cross-sectional online survey. We computed a variability index to measure the relative use of interpersonal compared to intrapersonal ER, called inter-vs-intrapersonal ER, by subtracting participants’ self-reported general tendency to use intrapersonal ER from their tendency to use interpersonal ER. The scores closer to 0 indicate more similarity in the use of interpersonal and intrapersonal ER and thus a higher variability. Larger positive or negative scores indicate lower variability; with positive scores indicating a preference for interpersonal ER, and negative scores indicating a preference for intrapersonal ER. The advantage of employing the subtraction method to calculate variability over the often-used standard deviation (SD) method [9,43] is its capability to not only quantify the extent but also the direction of variability. In other words, it not only shows the extent to which one strategy was preferred over the other, but also allows us to understand which ER strategy (interpersonal or intrapersonal) was preferred. This approach offers clear interpretability and is particularly relevant to our research question and hypothesis. Based on our hypotheses, we expected a positive association between attachment anxiety and inter-vs-intrapersonal ER (H1.1) and a negative association between attachment avoidance and inter-vs-intrapersonal ER (H1.2). For a visualization of hypotheses, see Figure 1. This study was pre-registered on the Open Science Framework (OSF; see https://osf.io/m2t9g, accessed on 1 February 2021; see Appendix A for deviations from pre-registration).

In order to limit the influence of confounding and other extraneous variables, we controlled for other potential variables that might influence inter-vs-intrapersonal ER. More specifically, we controlled for participants’ gender, relationship duration and relationship quality as they are shown to influence interpersonal processes in romantic relationships, particularly support seeking; for example, to regulate emotions [45]. Furthermore, the personality trait neuroticism is shown to be strongly associated with intrapersonal ER strategies such as suppressing and avoiding emotions, rumination on negative emotions, and isolating oneself [46]. Additionally, the personality trait extraversion is shown to be associated with support seeking from others [47]. Therefore, we also controlled for neuroticism and extraversion. Since the study was conducted during the spread of the COVID-19 pandemic, we also controlled for the extent to which participants’ responses were influenced by the COVID situation.

## 3. Materials and Methods of Study 1

### 3.1. Participants

The participants were included in the study if they were at least 18 years old, involved in a monogamous romantic relationship for at least six months, and in possession of a smartphone. To ensure independent data, only one partner of a couple could participate. Participants were not excluded based on physical or mental illness. Participants were recruited from April to October 2020, via the Radboud University participant recruitment system (SONA), and online advertisement on social media (e.g., Facebook, Instagram, WhatsApp). Based on a prior power analysis using G*Power 3.1.9.2 [48], for a multiple linear regression analysis, the sample should consist of at least 73 participants when using an estimated effect size of 0.15, significance level of 0.05, and a power of 0.90. Initially, 203 persons signed up for the study. The participants were excluded if they did not consent to participate (*n* = 3), did not complete the questionnaire (*n* = 23), reported that they did not answer the questionnaire seriously (*n* = 2), and did not meet the inclusion criteria concerning relationship length (*n* = 1). The final sample included 174 adults (133 female), aged between 18 and 58 (*M* = 23.79, *SD* = 7.63) with a relationship length between 6 months and 41 years (*M* = 3.5 years, *SD* = 6.1 years). The majority of participants were White/Caucasian (87%), unmarried (93%), not cohabiting with their partner (62%), and without children (95%). For details of the demographic characteristics of the participants, see Appendix A. The participants were compensated via study credit points (if applicable) and/or entering a lottery of 50 Euros.

### 3.2. Procedure

The study included an online survey (on Qualtrics) with a battery of questionnaires. Before proceeding with the questionnaires, participants received information about the study and were asked to give their consent. The survey took an average of 25 min to complete.

### 3.3. Measures

#### 3.3.1. Adult Attachment

The independent variables, adult attachment anxiety and avoidance, were measured using Experiences in Close Relationships-Revised (ECR-R) [49]. This scale consists of two subscales, attachment anxiety (18 items; *α* = 0.88) and attachment avoidance (18 items; *α* = 0.90). Participants indicated their level of agreement with each item on a Likert-type scale from 1 = strongly disagree to 7 = strongly agree. Attachment anxiety and attachment avoidance scores were computed by calculating the mean of participants’ responses on the corresponding items. Higher means indicate higher attachment anxiety or avoidance.

#### 3.3.2. Inter-vs-Intrapersonal Emotion Regulation

The dependent variable inter-vs-intrapersonal ER was defined as an index of variability, indicating the relative use of interpersonal compared to intrapersonal ER. Inter-vs-intrapersonal ER was measured using a 2-item self-constructed scale that asked participants about their general ER tendency when experiencing negative or distressing emotions (i.e., My general tendency is to: 1 item for interpersonal emotion regulation, “seek out the company or support of my romantic partner or other people close to me”; 1 item for intrapersonal emotion regulation: “try to handle the situation by myself”). The participants responded to the items on a Likert-type scale from 1 = almost never to 5 = almost always. To compute inter-vs-intrapersonal ER score, the interpersonal score minus intrapersonal score was calculated. The scores closer to 0 indicate more similarity in the use of interpersonal and intrapersonal ER and thus a higher variability between strategies. Larger positive or negative scores indicate lower variability. Positive scores indicate more use of interpersonal compared to intrapersonal ER. Negative scores indicate less use of interpersonal compared to intrapersonal ER.

#### 3.3.3. Relationship Quality

The control variable relationship quality was measured using a 18-item Perceived Relationship Quality Component (PRQC) [50]. An example item is “How satisfied are you with your relationship?”. Participants responded to each item on a Likert-type scale, from 1 = not at all to 7 = extremely. The relationship quality score was computed by calculating the mean of the participants’ responses on all items (*α* = 0.93).

#### 3.3.4. Personality Types

The control variables neuroticism and extraversion were measured using the 10-item Personality Inventory (TIPI) scale [51]. The TIPI scale comprises two items for each of five personality traits. Participants indicated their agreement with each item on a Likert-type scale from 1 = strongly disagree to 7 = strongly agree. Extraversion (*α* = 0.79) and neuroticism (*α* = 0.75) were computed by calculating the mean of the participants’ responses on the corresponding items.

#### 3.3.5. COVID

At the final stage of the survey, we asked participants to respond to a 1-item question “To what extent do you think your answers are different from a normal time when there was no Corona?” on a sliding bar ranging from 0 = not at all to 100 = very much. This self-developed variable was called COVID and was controlled for in the analysis.

### 3.4. Data Preparation and Analysis

All data preparation and analysis were performed in R version 4.0.3 [52]. Since all survey questions were forced response, we had no missing values. In participants’ responses to relationship length, some entries did not specify whether the indicated number referred to years or months. For these cases, we coded the entries as “NA” to denote missing information. To analyze the data, we ran a multiple linear regression using the function lmer [53] with attachment anxiety, attachment avoidance and their interaction term as independent variables and inter-vs-intrapersonal ER as the outcome variable. We also controlled for relationship length (in months), gender (dummy coded: 1 = female, 0 = male; named female), relationship quality, extraversion, neuroticism, and COVID. All continuous variables were centered around the sample mean scores before entering in the analysis. Notably, the interaction between attachment anxiety and avoidance was included in the analysis following Fraley’s [54] advice. Additionally, the initial model included age as a control variable; however, due to collinearity between age and relationship length, and since the model with relationship length (and not with age) could better capture the variance in the dependent variable than the model with age (and not relationship length), age was excluded from the final model.

## 4. Results of Study 1

### 4.1. Preliminary Analysis of Data

Descriptive statistics are presented in Table 1. Notably, the mean score for inter-vs-intrapersonal ER (*M* = −0.01) indicates that, on average, individuals exhibited an equal inclination toward using interpersonal and intrapersonal ER, with a slightly higher tendency toward intrapersonal ER. Participants reported varying levels of attachment insecurity, with mean levels falling in the lower to mid-range of the scale. On average, participants reported little influence of COVID on their responses.

Correlations between variables are presented in Appendix A. Attachment anxiety and avoidance were positively correlated (*r* = 0.41, *p* < 0.001). Inter-vs-intrapersonal ER was significantly negatively correlated with attachment avoidance (*r* = −0.39, *p* < 0.001) but not significantly with attachment anxiety (*r* = −0.13, *p* = 0.071). Furthermore, higher inter-vs-intrapersonal ER was reported by females (*r* = 0.29, *p* < 0.001), participants with higher relationship quality (*r* = 0.19, *p* = 0.013), and those with lower neuroticism (*r* = −0.27, *p* < 0.001). Age and relationship length were highly correlated (*r* = 0.80, *p* < 0.001).

### 4.2. Main Analysis

The model explained 21.8% of the variance in inter-vs-intrapersonal ER (*F*(9, 159) = 6.204, *p* < 0.001). As can be seen in Table 2, the effect of attachment anxiety was not significant, meaning that unlike H1.1, the relative use of interpersonal compared to intrapersonal ER did not depend on one’s level of attachment anxiety. Attachment avoidance was significantly negatively associated with inter-vs-intrapersonal ER (H1.2), meaning that individuals with higher attachment avoidance reported a lower reliance on interpersonal compared to intrapersonal ER when dealing with their negative emotions when stressed. The results also indicated that the variability between strategies decreased at higher scores of attachment avoidance with an increased preference for intrapersonal compared to interpersonal ER (see Figure 2). Attachment anxiety and attachment avoidance did not significantly interact in influencing inter-vs-intrapersonal ER, suggesting that the effect of attachment avoidance on the relative use of interpersonal compared to intrapersonal ER did not depend on different levels of attachment anxiety.

### 4.3. Exploratory Analysis

To attain further insight into the association between attachment and the measure of interpersonal relative to intrapersonal ER, we performed two other exploratory regression analyses with interpersonal ER and intrapersonal ER separately. The results (see Appendix A) showed that attachment anxiety was not significantly associated with interpersonal ER (*b* = −0.06, *SE* = 0.10, *p* = 0.542) or intrapersonal ER (*b* = 0.07, *SE* = 0.12, *p* = 0.559). In contrast, higher attachment avoidance was associated with both a lower tendency to use interpersonal ER (*b* = −0.61, *SE* = 0.12, *p* < 0.001) and a higher tendency to use intrapersonal ER (*b* = 0.33, *SE* = 0.14, *p* = 0.023). That is, individuals with higher attachment avoidance reported less proximity and support seeking and more self-regulation compared to those with lower attachment avoidance. We did not find significant interactions between the two attachment orientations for interpersonal (*b* = −0.05, *SE* = 0.09, *p* = 0.577), nor for intrapersonal (*b* = −0.03, *SE* = 0.11, *p* = 0.774).

## 5. Discussion of Study 1

As a brief discussion of results of the Study 1, attachment anxiety had no influence on the amount of using interpersonal versus intrapersonal ER, providing no support for our first hypothesis that people with higher attachment anxiety have lower ER variability with an over-reliance on interpersonal versus intrapersonal ER (H1.1). However, the results supported our second hypothesis (H1.2), as higher attachment avoidance was associated with lower interpersonal compared to intrapersonal ER, indicating a lower variability in higher levels of attachment avoidance. In other terms, individuals with higher (compared to lower) attachment avoidance had a higher tendency to rely on intrapersonal ER strategies over seeking proximity and support from close others. These findings provide support for overreliance on one particular ER strategy for avoidantly attached individuals, but not for anxiously attached individuals.

## 6. Study 2

In this study, we combined a baseline online survey with an ESM study for a period of one week. This study was pre-registered on the OSF (see https://osf.io/et5a2, accessed on 1 February 2021; see Appendix A for deviations from pre-registration). Our first aim was to replicate the results of the first study regarding the association between attachment and the relative use of interpersonal compared to intrapersonal ER in both baseline data and ESM data. In the baseline survey, similar to Study 1, we calculated an index of ER variability, indicating the relative use of interpersonal compared to intrapersonal ER, called inter-vs-intrapersonal ER. This index was calculated as a self-report on the extent to which people generally tend to use interpersonal ER minus the extent to which they generally tend to use intrapersonal ER when they experience stress.

Similarly, in ESM, we calculated momentary inter-vs-intrapersonal *ER* as momentary self-report on the extent to which individuals used interpersonal ER minus the extent to which they used intrapersonal ER when they experienced stressful events in daily life. We particularly asked for stressful life events since it is the focus of our research. Based on our hypotheses outlined in the Introduction, in both baseline and ESM data, we tested for a positive association between attachment anxiety and inter-vs-intrapersonal ER (H1.1) and a negative association between attachment avoidance and inter-vs-intrapersonal ER (H1.2).

In addition, as a second indicator of variability, we calculated the extent to which inter-vs-intrapersonal ER varied over time. Following Aldao et al.’s [9] advice and similar to Blanke et al. [43], we calculated the standard deviation of all momentary assessments of inter-vs-intrapersonal ER for each participant. In concrete terms, a higher standard deviation reflects higher variation in the relative use of interpersonal compared to intrapersonal ER. We expected that people with higher attachment anxiety or avoidance will show lower standard deviations (H2.1, H2.2), indicating lower variability in inter-vs-intrapersonal ER.

Finally, we investigated the association between attachment and ER flexibility; that is, the variable use of ER strategies depending on the specific context, and in this case, partner availability. We used Aldao et al.’s [9] suggestion to calculate a coefficient that measures the extent to which ER variability and changes in the context/environment covary together, where larger coefficients represent higher ER flexibility. We performed a regression analysis to compute the extent to which inter-vs-intrapersonal ER (as an index of ER strategy variability) depends on changes in partner availability. Overall, we expected that individuals would use more interpersonal compared to intrapersonal ER when their romantic partner is available compared to when the partner is unavailable (H3.1). More importantly, we tested whether attachment anxiety and avoidance moderated the association between partner availability and inter-vs-intrapersonal ER. Specifically, we expected the association between partner availability and use of inter-vs-intrapersonal ER to be weaker for those with higher (compared to lower) attachment anxiety and avoidance (H3.2, and H3.3). For a visualization of hypotheses, see Figure 1.

## 7. Materials and Methods of Study 2

### 7.1. Participants

As stated before, the study included a baseline survey followed by a 1-week ESM procedure. In our baseline data analysis, we aimed to gather information from a minimum of 110 participants. This decision was guided by an a priori power analysis, considering an estimated effect size (f2) of 0.07 (the smallest expected effect size based on the results of the Study 1), a significance level of 0.05, and a power of 0.80, calculated using G*Power 3.1.9.2 [48]. For the ESM data, where we collected multiple observations per participant and employed a multi-level model with enhanced power, we expected to have sufficient statistical power with a minimum of 110 participants. The inclusion criteria were being over 18 years old, involved in a monogamous, romantic relationship for at least three months, and in possession of a smartphone. To ensure independent data, only one partner of a couple could participate. Participants were not excluded based on physical or mental illness. Participants were recruited from April 2021 to March 2022, via the Radboud University participant recruitment system (SONA), and online advertisement on social media (e.g., Facebook, Instagram, WhatsApp). A portion (but not all) of participants who completed the baseline survey also completed the ESM procedure. The participants were compensated via study credit points (if applicable) for completing either the baseline or both the baseline and ESM procedures. Furthermore, all participants who completed the baseline questionnaire and the ESM procedure with at least a 75% compliance rate entered a lottery (4 × 50€ cash prizes).

#### 7.1.1. Baseline

In total, 435 persons signed up for the study. We excluded participants who did not consent to participate (*n* = 74), did not complete the questionnaire (*n* = 65), reported that they did not answer the questionnaire seriously (*n* = 4), and did not meet the inclusion criteria concerning minimum relationship length (*n* = 5). The final sample included 287 adults (244 female, 3 other), aged between 18 and 58 (*M* = 22.13, *SD* = 5.04) with a relationship length between 3 months and 21 years (*M* = 2.5 years, *SD* = 2.5 years). Compared to Study 1, Study 2 baseline participants were younger (*t*(265.84) = −2.56, *p* = 0.011). Similar to Study 1, the majority of participants were White/Caucasian (87%), unmarried (96%), not cohabiting with their partner (72%), and without children (96%). For details of the demographic characteristics of the participants, see Appendix A.

#### 7.1.2. Experience Sampling Method

Initially, 136 eligible participants signed up for the ESM part of the study. We excluded data of those who dropped out during the ESM part (*n* = 9) and did not meet the commonly applied pre-defined 33% compliance rate (*n* = 3). The final ESM analyses included 124 participants (104 female) aged between 18 and 58 years old (*M* = 22.45, *SD* = 6.39 years) with a relationship duration between 3 months and 21 years (*M* = 2.81 years, *SD* = 3.5 years) in the ESM analysis. Other demographic characteristics were similar to those of Study 1 and Study 2 baseline participants (for details, see Appendix A).

### 7.2. Procedure

The study included two main parts: (1) an online baseline survey (on Qualtrics) and (2) an ESM procedure, using the SEMA^3^ app SEMA3: Smartphone Ecological Momentary Assessment, version 1.4.2(48) [55]. In the baseline survey (first part), participants responded to a battery of questionnaires. Before proceeding with the questionnaires, participants received information about the study and were asked to give their consent. The survey took an average of 25 min to complete. Upon completion of the baseline survey, participants were instructed to schedule an individual online meeting (i.e., briefing session) with the researchers. During the briefing session, participants were guided through the installation and usage of the SEMA3 app on their smartphones and the ESM procedure using a pre-established protocol and an informative PowerPoint presentation.

For each participant, a 7-day ESM procedure started on the next day after their briefing session. Each day, on their smartphones, participants received seven semi-randomly timed prompts throughout the day (one prompt randomly programmed within each 2 h interval within 14 awake hours) and one evening prompt in the evening. The schedule of prompts was adjusted individually to each participant’s usual daily schedule so that the prompts would occur when the participant was usually awake.

The daily prompts had to be responded within 30 min; otherwise, the questionnaire would be missed. The daily prompts contained momentary questionnaires which took around two to three minutes to complete. In each questionnaire, participants first reported their momentary emotions. Then, they were instructed to think of a situation when they felt bad/stressed since the last notification. This was followed by questions about the context of the event and the emotion regulation strategies that the participant used to deal with their stress since the last notification. Note that in the briefing session, the participants were instructed to report any major or minor stressful events (e.g., missing a bus). Nevertheless, participants had the option to report that no stressful events occurred since the last prompt, which resulted in receiving a set of filler questions that were very similar to the actual questions in content and number but were not of interest to the current study. Furthermore, participants received a 4 min evening questionnaire on their average ER during each day which was not included in the data analysis of the current study. By the end of the seventh day, upon the request of participants, up to two extra days of ESM procedure were provided to increase the compliance rate. Eleven participants used this opportunity to increase their compliance rate, aiming to qualify for full study credits and/or gaining entry into the lottery. Upon completion, participants received a debriefing via e-mail and access to their data summary provided by the SEMA3 app.

### 7.3. Measures

The baseline measures were the same as in Study 1, with two differences: Firstly, to measure adult attachment, we used a shorter version of ECR-R (ECR-R-General Short Form) [56] with 10 items for attachment anxiety (*α* = 0.87) and 10 items for attachment avoidance (*α* = 0.84). Additionally, to measure relationship quality, we used only three subscales (trust, commitment, and satisfaction) from the PRQC [50] including 9 items (*α* = 0.90). Notably, the Cronbach’s alpha for personality traits neuroticism and extraversion were *α* = 0.63 and *α* = 0.73, respectively. In the ESM part, if the participant indicated that they experienced a bad/stressful event, they were asked to respond to a series of questions about that event, including in the presented order:

#### 7.3.1. Partner Availability

The participants responded to a multi-select multiple choice question “When I felt bad/stressed, I was in contact with …”, with 10 response items (i.e., Nobody, Romantic partner, Housemates, Family, Family (living elsewhere), Colleagues/Classmates, Friend, Acquaintances, Strangers, Other). The participants were previously informed that being in contact means, for example, face-to-face, via phone call, or online contact. The variable partner availability was dummy coded as 1 if the participant indicated that they were in contact with their partner and as 0 if otherwise. The variable others availability was dummy coded as 1 if the participant indicated that they were with anyone else than their romantic partner and 0 as otherwise.

#### 7.3.2. Stressfulness of the Event

The participants responded to a 1-item question “To what extent did you feel bad/stressed?” on a sliding bar ranging from 0 = not at all to 10 = very much.

#### 7.3.3. Momentary Inter-vs-Intrapersonal Emotion Regulation

The participants responded to “What did you do to deal with your negative feelings/stress” on 3 items (1 item for interpersonal: “I reached out for the company and support of my partner.” 1 item for intrapersonal: “I tried to handle my negative feelings/stress by myself.” 1 item “I reached out for the company and support of others”) on a visual sliding bar from 0 = not at all to 10 = very much. As the first indicator of variability, indicating the relative use of interpersonal compared to intrapersonal ER in each moment, momentary inter-vs-intrapersonal ER was calculated as the interpersonal score minus intrapersonal score for each notification (we left out the scores on the third item, measuring interpersonal ER using others than romantic partner). A score of 0 indicates equal use of interpersonal compared to intrapersonal ER, with scores closer to 0 indicating higher variability and larger positive or negative scores indicating lower variability. Positive scores indicate more use of interpersonal compared to intrapersonal ER. Negative scores indicate less use of interpersonal compared to intrapersonal ER. As the second indicator of ER variability, the variation in the relative use of interpersonal compared to intrapersonal ER over time, the SD of momentary inter-vs-intrapersonal ER was calculated across all momentary assessments for each participant based on previous research [9,43].

All ESM measures were developed based on items found in the ESM Item Repository [57].

### 7.4. Data Preparation and Analysis

All data preparation and analysis were performed in R, version 4.0.3 [52]. The baseline measures from Qualtrics and ESM measures from SEMA3 were downloaded as separate csv files, and then imported in R, where data were preprocessed, and variable scores were calculated (for details of pre-processing steps, see Appendix A). The analysis of the baseline data was equivalent to the one reported in Study 1. In all analysis, in order to limit the influence of confounding and other extraneous variables, and since we aimed to replicate the results of Study 1, we used the same control variables as Study 1, namely, relationship length, gender, relationship quality, extraversion, neuroticism, and the effect of COVID. Furthermore, in ESM analyses, we controlled for momentary stressfulness of the event, availability of the partner, and availability of others than partner. Notably, since individuals can use interpersonal ER not only with their romantic partner but also with others, we considered controlling for the variable interpersonal ER with others than partner. However, this variable is correlated with and affected by our control variable availability of others than partner and, it is challenging to determine the sequence of events—whether interpersonal ER occurred with the partner or others first. Therefore, for the sake of model parsimony and to avoid potential issues related to reverse causality and multicollinearity, we opted not to include interpersonal ER with others.

#### 7.4.1. Main Analyses

For the analysis of the ESM data to answer RQ1 part 1 (concerning the relative use of interpersonal compared to intrapersonal ER) and RQ2, we performed two multilevel models using the lme function from R package nlme [58] and with momentary inter-vs-intrapersonal ER as an outcome variable and attachment anxiety, attachment avoidance, and their interaction term as independent variables. For both models, time-invariant control variables—including relationship length, gender (female), relationship quality, extraversion, neuroticism, and COVID—as well as time-variant control variable time were included as fixed effects. Furthermore, time-variant control variables such as stressfulness of event, partner availability, and others availability were added as both fixed and random effects. Additionally, for RQ2, the two-way and three-way interactions of partner availability with attachment anxiety and attachment avoidance were entered as fixed effects. Following Bolger and Laurenceau’s [59] advice, all continuous independent time-invariant variables were grand-mean centered around the population mean. All continuous independent time-variant variables were person-mean centered around participant means.

To answer RQ2 part 2 (the variation in the relative use of interpersonal compared to intrapersonal ER over time), a multiple linear regression analysis was performed with SD of momentary inter-vs-intrapersonal ER as the outcome variable, and attachment anxiety, attachment avoidance, and their interaction term as independent variables. In the initial phase of data analysis, a set of control variables, namely relationship length, female, relationship quality, extraversion, neuroticism, and COVID, were included in the model. However, the model did not significantly explain the variance in the dependent variable. To refine the model, control variables were systematically removed one by one, starting with those exhibiting the lowest correlation with the outcome variable. Following the sequential removal of extraversion and female, the model achieved statistical significance. Therefore, the final model included centered values of relationship length, relationship quality, neuroticism, and COVID as control variables. Notably, the patterns of effect size and significance did not change with removal of extraversion and female. Additionally, we computed mean levels of stressfulness of events, partner availability, and others availability for each participant and added it to the model as control variables.

#### 7.4.2. Exploratory Analysis

For each model in the main analyses, two additional exploratory analyses were performed, maintaining the main analysis model’s independent variables but using different outcome variables: one with interpersonal ER as the outcome and another with intrapersonal ER.

## 8. Results of Study 2

### 8.1. Preliminary Analyses

Participants completed a total of 6052 ESM notifications with an average 38 out of 49 responses, yielding a mean compliance rate of 78.50% (*SD* = 14). Participants reported experiencing a stressful event in 2677 notifications, averaging 22 notifications per participant, meaning a mean 57.90% (*SD* = 24) of responded notifications. Participants reported their partner was available in a mean 32% (*SD* = 23) of the stressful events. Descriptive statistics are presented in Table 1. Notably, compared to Study 1, Study 2 baseline had a higher relationship quality, *t*(363.08) = 2.08, *p* = 0.038, reported an average higher attachment avoidance, *t*(352.92) = 2.12, *p* = 0.035, and indicated a higher influence of COVID on their responses *t*(374.85) = 2.86, *p* = 0.005. Participants of Study 2 baseline did not differ from Study 1 participants in their average attachment anxiety, *t*(416.64) = 0.78, *p* = 0.439, and inter-vs-intrapersonal ER, *t*(372.57) = 1.37, *p* = 0.172. Study 2 ESM did not differ from the larger sample of Study 2 baseline in average scores of baseline variables (e.g., attachment anxiety; see Appendix A).

Correlations between Study 2 baseline variables are presented in the Appendix A. Attachment anxiety and avoidance were positively correlated (*r* = 0.30, *p* < 0.001). Similar to Study 1, inter-vs-intrapersonal ER was negatively correlated with attachment avoidance (*r* = −0.42, *p* < 0.001) but not correlated with attachment anxiety (*r* = −0.05, *p* = 0.467). That is, individuals with higher attachment avoidance had a lower tendency to use inter-vs-intrapersonal ER.

Correlations between Study 2 ESM variables are presented in the Appendix A. Notably, baseline and ESM measures of inter-vs-intrapersonal ER were only moderately positively correlated (*r* = 0.31, *p* < 0.001). Nevertheless, similar to the baseline measure, ESM measure of inter-vs-intrapersonal ER was negatively correlated with attachment avoidance (*r* = −0.29, *p* < 0.001) but not with attachment anxiety (*r* = −0.09, *p* = 0.283). Inter-vs-intrapersonal ER was also positively correlated with relationship quality (*r* = 0.19, *p* = 0.019).

### 8.2. Emotion Regulation Variability

#### 8.2.1. Inter-vs-Intrapersonal Emotion Regulation: Baseline Cross-Sectional Data

##### Main Analysis

The model explained 21.8% of the variance in inter-vs-intrapersonal ER (*F*(10, 271) = 8.812, *p* < 0.001), as an index of ER strategy variability, indicating the relative use of interpersonal compared to intrapersonal ER. The results are presented in Table 2. As in Study 1, the effect of attachment anxiety was not significant, meaning that the relative use of interpersonal compared to intrapersonal ER did not depend on one’s level of attachment anxiety, not supporting H1.1. Replicating the findings of Study 1, the association with attachment avoidance was negative and significant, supporting H1.2; that is, on average, the higher one’s level of attachment avoidance, the less they reported to use interpersonal compared to intrapersonal ER—or put differently, the more they relied on intrapersonal compared to interpersonal ER. The results indicated a lower ER strategy variability for people with high levels of attachment avoidance (see Figure 2). Attachment anxiety and attachment avoidance did not significantly interact in influencing inter-vs-intrapersonal ER, suggesting that the effect of attachment avoidance on inter-vs-intrapersonal ER did not depend on different levels of attachment anxiety.

##### Exploratory Analysis

The results (see Appendix A) replicated the findings of Study 1. Attachment anxiety was not significantly associated with neither interpersonal ER (*b* = 0.07, *SE* = 0.07, *p* = 0.290) nor intrapersonal ER (*b* = 0.06, *SE* = 0.07, *p* = 0.438). In contrast, on average, an increase in attachment avoidance was associated with both a lower tendency to use interpersonal ER (*b* = −0.62, *SE* = 0.08, *p* < 0.001) and a higher tendency to use intrapersonal ER (*b* = 0.48, *SE* = 0.08, *p* < 0.001). The association between attachment avoidance and interpersonal or intrapersonal ER did not depend on the levels of attachment anxiety, as indicated by the insignificant interactions between the two attachment orientations (for interpersonal *b* = 0.09, *SE* = 0.06, *p* = 0.151; for intrapersonal *b* = −0.06, *SE* = 0.07, *p* = 0.331).

#### 8.2.2. Inter-vs-Intrapersonal Emotion Regulation: Experience Sampling Method Data

##### Main Analysis

The results are reported in Table 2. The results replicated our previous results of the Study1 and Study 2 baseline analyses. Using our first indicator of variability (the relative use of interpersonal vs. intrapersonal ER), the effect of attachment anxiety was not significant, providing no support for H1.1; that is, the momentary relative use of interpersonal compared to intrapersonal ER did not depend on one’s level of attachment anxiety. The effect of attachment avoidance was significant, supporting H1.2; that is, on average, higher levels of attachment avoidance were associated with less use of interpersonal compared to intrapersonal ER on average across all momentary assessments. The results indicated a lower ER strategy variability for those with higher attachment avoidance (see Figure 2). In contrast to the baseline data, the ESM results showed a significant interaction effect of attachment anxiety and avoidance in predicting momentary inter-vs-intrapersonal ER. Plotting the interaction effect showed that the effect of attachment avoidance on the relative use of interpersonal compared to intrapersonal ER decreased as the level of attachment anxiety increased (see Figure 3). The analysis of the simple slopes showed that on average, higher attachment avoidance was associated with less use of interpersonal compared to intrapersonal ER when attachment anxiety was low (−1SD; *b* = −1.26, *p* = 0.002) or average (*b* = −0.77, *p* = 0.015). When attachment anxiety was high (+1SD), more avoidance was not significantly associated with momentary relative use of interpersonal compared to intrapersonal ER (*b* = −0.28, *p* = 0.453). Thus, participants higher in avoidance showed a lower use of interpersonal compared to intrapersonal ER only when their attachment anxiety was low or medium.

##### Exploratory Analysis

The results (see Appendix A) replicated the findings of the Study 1 and Study 2 baseline, showing no effect of attachment anxiety on neither interpersonal ER (*b* = 0.12, *SE* = 0.16, *p* = 0.496) nor intrapersonal ER (*b* = 0.12, *SE* = 0.16, *p* = 0.465). In contrast to attachment anxiety, on average, an increase in attachment avoidance was associated with a lower momentary use of interpersonal ER (*b* = −0.66, *SE* = 0.20, *p* = 0.001), replicating our earlier findings; this association did not depend on the levels of attachment anxiety, as indicated by the insignificant interactions between the two attachment orientations (*b* = 0.15, *SE* = 0.15, *p* = 0.311). Different from our earlier findings, attachment avoidance was not associated with the momentary use of intrapersonal ER (*b* = 0.17, *SE* = 0.20, *p* = 0.408). However, there was a significant interaction between the two attachment orientations (*b* = −0.31, *SE* = 0.15, *p* = 0.035). Exploring the interaction effect showed that the association between attachment avoidance and intrapersonal ER was positive when attachment anxiety was low (*b* = 0.48, *z* = 1.81, *p* = 0.070) or average (*b* = 0.17, *z* = 0.83, *p* = 0.404), and negative when attachment anxiety was high (*b* = −0.14, *z* = −0.60, *p* = 0.548). None of these associations were, however, significant.

#### 8.2.3. Standard Deviation of Inter-vs-Intrapersonal Emotion Regulation

##### Main Analysis

The results are presented in Table 3. The model explained 7.2% of the variance in SD of inter-vs-intrapersonal ER (*F*(10, 112) = 1.944, *p* = 0.046). Supporting H2.1, higher attachment anxiety was significantly associated with a lower SD in the relative use of interpersonal compared to intrapersonal ER among people high in attachment anxiety, indicating lower variability. Providing no support for H2.2, the effect of attachment avoidance on SD was not significant, indicating that the variability in the relative use of interpersonal compared to intrapersonal ER did not depend on one’s level of attachment avoidance. Furthermore, attachment anxiety and attachment avoidance did not significantly interact in influencing SD; that is the association between attachment anxiety and the variation in the relative use of interpersonal compared to intrapersonal ER did not differ across different levels of attachment avoidance.

##### Exploratory Analysis

The variance in the SD of *interpersonal* ER was not significantly explained by neither the model (*R^2^* adjusted = 0.004, *F*(10, 112) = 1.055, *p* = 0.403), nor attachment anxiety (*b* = −0.13, *SE* = 0.11, *p* = 0.222), attachment avoidance (*b* = 0.04, *SE* = 0.14, *p* = 0.756), or the interaction between them (*b* = 0.04, *SE* = 0.10, *p* = 0.718). However, 9.1 percent of variance in SD of *intrapersonal* ER was explained by the model (*F*(10, 112) = 2.222, *p* = 0.021). Higher attachment anxiety was associated with lower SD of intrapersonal ER (*b* = −0.23, *SE* = 0.10, *p* = 0.022). That is, on average, people with higher attachment anxiety showed less variability in the amount of self-regulation they used during a week. For details of the results, see Appendix A. Attachment avoidance was not associated with SD of intrapersonal ER (*b* = 0.07, *SE* = 0.13, *p* = 0.613). The association between attachment anxiety and SD of intrapersonal ER was moderated by attachment avoidance (*b* = 0.24, *SE* = 0.09, *p* = 0.012), as it was larger when attachment avoidance was low (*b* = −0.42, *z* = −1.04, *p* = 0.300) compared to when it was high (*b* = −0.05, *z* = −0.12, *p* = 0.905). None of the simple slopes were, however, significant.

### 8.3. Emotion Regulation Flexibility

#### 8.3.1. Main Analysis

The results are presented in Table 4. Partner availability (versus partner absence) predicted higher inter-vs-intrapersonal ER as a main effect; that is, when their partner was available, people used more interpersonal compared to intrapersonal ER than when their partner was not available (H3.1). This indicates a general flexibility in the use of interpersonal compared to intrapersonal ER with respect to the contextual factor of partner availability. The results, however, showed no significant interaction between partner availability and neither attachment anxiety (H3.2) nor attachment avoidance (H3.3), indicating that the level of flexibility was not influenced by attachment orientations.

#### 8.3.2. Exploratory Analysis

The results (see Appendix A) revealed that participants used higher interpersonal ER when their partner was available (versus absent; *b* = 4.03, *SE =* 0.23, *p* = <0.001). This association was moderated by attachment anxiety (*b* = −0.51, *SE =* 0.23, *p* = 0.030). That is, as expected, the association between partner availability and interpersonal ER was weaker for those with high attachment anxiety (*b* = 3.53, *t* = 10.67, *p* < 0.001) than for those with low anxiety (*b* = 4.54, *t* = 13.95, *p* < 0.001; see Figure 4). This suggests lower flexibility in the use of interpersonal ER for people with higher attachment anxiety. Partner availability was negatively associated with intrapersonal ER (*b* = −1.92, *SE =* 0.18, *p* < 0.001). That is, when their partner was available (compared to absent), participants used less self-regulation. This association was, however, not moderated by neither attachment anxiety (*b* = 0.10, *SE* = 0.18, *p* = 0.573) nor attachment avoidance (*b* = 0.02, *SE* = 0.23, *p* = 0.943). That is, neither attachment anxiety nor avoidance were associated with lower flexibility in self-regulation.

## 9. General Discussion

We investigated the association between adult attachment to a romantic partner and the extent to which one flexibly uses interpersonal compared to intrapersonal ER when regulating their own emotional response to daily life stressful situations. Specifically, we measured the extent to which one (1) variably uses different ER strategies, and (2) flexibly varies the use of ER strategies depending on the context (in this case: whether the partner was available or not). Before discussing the implications, we summarize the various results of our studies and clarify their conceptual meaning.

Across two studies and three different types of analyses (based on two cross-sectional data and one ESM data), our results consistently suggested that in contrast to what we hypothesized (H1.1), individuals higher (compared to lower) in attachment anxiety did not show a stronger preference for interpersonal compared to intrapersonal ER. However, as hypothesized (H2.1), they showed a lower variability (SD) in the relative use of interpersonal compared to intrapersonal ER across different occasions over time. Therefore, their relative use of interpersonal compared to intrapersonal ER did not vary from individuals with lower attachment anxiety on average; however, it did vary or fluctuate significantly less across different occasions. As hypothesized (H1.2), participants higher (compared to lower) in attachment avoidance reported a stronger preference for self-regulation (intrapersonal ER) compared to relying on others (interpersonal ER), indicating lower variability in ER. However, we found no support that higher attachment avoidance was associated with lower variability when looking at the standard deviations of using inter- vs. intrapersonal ER strategies (H2.2). This indicates that, while high (versus low) avoidant people across situations, on average, tend to have a strong preference for intra- over interpersonal ER, high avoidant people are ‘variable’ in the sense that they range from either having a slight to having a strong preference for intra- over interpersonal ER across occasions.

Finally, as expected (H3.1), in general, individuals’ relative use of interpersonal compared to intrapersonal ER increased when the partner was available compared to when they were absent, indicating ER flexibility. We found partial (only in our exploratory analysis) support for our hypothesis that people with higher attachment anxiety show lower flexibility (H3.2). That is, while individuals used higher levels of interpersonal ER when their partner was available (compared to absent), this association between partner availability and interpersonal ER was weaker for those with higher (compared to lower) attachment anxiety. Specifically, when their partner was available, they used slightly less interpersonal ER; and when their partner was not available, they used more interpersonal ER compared to those with lower attachment anxiety. We found no support for our hypothesis that people with higher attachment avoidance show lower flexibility (H3.3); that is, similar to secure people, their relative use of interpersonal compared to intrapersonal ER increased when the partner was available compared to when they were absent. Together, the present findings make a novel contribution to the current literature on attachment and ER by providing initial insights into the role of attachment anxiety and avoidance in the ER flexibility.

### 9.1. Attachment Anxiety and Emotion Regulation Flexibility

Our results showed, both in the cross-sectional data as well as in the ESM in daily life, that individuals high (compared to low) in attachment anxiety did not show a stronger, or weaker, preference for interpersonal compared to intrapersonal ER. This was surprising as previous theorizing has often considered both emotional distress-amplification as well as heightened behavioral proximity and support seeking as indicators of hyper-activation that is typical of anxiously attached individuals [5,29,31,33,60]. Our findings raise the interesting theoretical question whether attachment anxiety may be characterized mainly by emotional and cognitive hyper-activation in response to threat (e.g., stress, rumination), but that anxious attachment is not typically characterized by interpersonal proximity and support seeking in terms of emotional regulation strategy.

The fact that people high in anxiety displayed a relatively low variability as indicated by lower standard deviations across occasions (and particularly low variability in the use of intrapersonal ER as the result of the exploratory analysis showed), may indicate less flexibility in their ER in response to changes in the environment (in this case, partner availability). Indeed, our exploratory results showed that individuals with higher attachment anxiety showed lower ER flexibility when considering interpersonal ER. That is, their amount of proximity and support seeking was less dependent on their partner availability compared to those with lower attachment anxiety. In John’s example, his support-seeking behavior was less covarying with Mary’s availability compared to a person who is less anxiously attached to their partner.

One explanation for our results on variability and flexibility can be the heightened sensitivity of highly anxious people to signs of threat [3,60], which perhaps makes it difficult to accurately perceive and respond to social cues. The ability to accurately evaluate a situation and perceive its contextual demands and opportunities is indeed a first step in having higher ER flexibility [11,42]. Less sensitivity to situational circumstances could lead individuals to gravitate toward certain emotion regulation strategies, regardless of the specific demands of a situation [9]. In John’s example, if his perception of Mary’s ability to be available is not accurate, he might engage in hyperactivating behavior by exaggerating his support seeking when Mary does not respond to his phone call. Another related explanation could be that the partner’s unavailability in times of stress may be appraised as particularly stressful by anxiously attached individuals [3,5,60,61], resulting in more support-seeking behaviors (i.e., more interpersonal ER). For example, if Mary does not respond to John’s phone call, his stress levels may rise, resulting in higher support seeking.

### 9.2. Attachment Avoidance and Emotion Regulation Flexibility

Individuals with higher attachment avoidance showed low variability between strategies in that, on average, they displayed a higher preference for the use of intra- over interpersonal ER. This aligns with previous research indicating that higher attachment avoidance correlates with deactivating strategies [5] such as increased autonomy [60,62], decreased interdependence with close others [63], and less support seeking from them [31,36,45]. Our study extends previous findings by simultaneously examining interpersonal and intrapersonal ER in individuals’ actual daily lives, with the ESM study providing real-life momentary evidence that highly avoidant individuals indeed tend to prioritize intrapersonal ER strategies over interpersonal ER when coping with daily life stressors.

While, on average, individuals with higher attachment avoidance showed a higher preference for intrapersonal compared to interpersonal ER, they did show variability as the strength of their preference varied largely across situations. That is, they showed a relatively large fluctuation across situations from having only a slightly to a strongly preferred use of intrapersonal ER over interpersonal ER. This finding seems consistent in that they flexibly used different amounts of interpersonal compared to intrapersonal ER depending on their partner’s availability. A possible theoretical implication of these observed patterns in our study is that the general lower tendency to use interpersonal compared to intrapersonal ER in avoidant individuals is not a matter of limited repertoire or capability; instead, it is likely that motivational factors and avoidant individuals’ perception of the usefulness of relying on others play a role. While they may be capable of using both inter- as well as intrapersonal ER strategies, they simply have a strong preference for intrapersonal ER. Indeed, in our study, similar to previous research [64], participants with higher attachment avoidance rated interpersonal ER as less helpful than those with lower avoidance (see exploratory correlations in Appendix A). This also aligns with other previous research where avoidant individuals reported a lower need for support from a romantic partner [65] and a preference for independent problem-solving and avoiding social interaction during distress [66,67].

Finally, analyses of our momentary data revealed that individuals with higher attachment avoidance, across occasions, relied more on intrapersonal compared to interpersonal ER particularly when their attachment anxiety levels were low or average. These results align with prior research, where individuals exhibiting both high avoidance and anxiety are classified as fearful avoidant [20], demonstrating inconsistent behaviors [68]. Our exploratory analyses (Appendix A) suggest that higher attachment anxiety may moderate the effect of attachment avoidance on intrapersonal ER. That is, similar to the explanation of Simpson and Rholes [68], the desire of fearful individuals for self-reliance might be buffered by a simultaneous inclination towards interdependence, resulting in ER patterns comparable to those of securely attached individuals in our data. This result shows the importance of looking at interpersonal and intrapersonal ER both individually and simultaneously.

### 9.3. Implications for Research and Practice

Our results indicate that both attachment anxiety and attachment avoidance are distinctively associated with indicators of limited ER flexibility, implying that lower ER flexibility may partly explain the ER difficulties of individuals with higher attachment insecurity. Specifically, attachment anxiety was associated with less variable and adaptive ER behavior in response to environmental changes, implying less accurate evaluation of situational demands. In contrast, our results imply that individuals with higher attachment avoidance are able to accurately detect environmental changes but exhibit inflexibility due to a general reluctance to rely on others, likely rooted in a predetermined expectation that others are unreliable for ER. This underscores the need for future research to explore the distinct mechanisms linking attachment insecurities with ER (flexibility).

On a more general and theoretical note, our findings reinforce the distinction between attachment anxiety and avoidance, confirming that they are distinct subsystems within the attachment system, rather than two sides of the same dimension. This distinction is crucial for developing targeted interventions and furthering theoretical frameworks in attachment and emotion regulation research.

Our results can also have important applied implications. Our findings imply that interventions for those with attachment insecurity should focus on increasing ER flexibility. For individuals with higher attachment anxiety, interventions could aim to enhance the ability to accurately recognize the situational demands and to use a variety of strategies accordingly. For those with higher attachment avoidance, interventions might focus on enhancing their willingness to rely on others, thereby improving their ability to use interpersonal ER.

### 9.4. Limitations and Directions for Future Research

Our findings should be interpreted in light of several limitations. First, our study focused on examining attachment-related ER during periods of stress, but our approach lacks insight into participants’ ER in positive life situations. Future research could explore attachment-related ER differences between positive and negative life events by incorporating both types of events concurrently in data collection and analysis. Second, in the ESM data, there was an uneven distribution of responses, with some individuals reporting only a few instances of stressful events, while others reported more frequent occurrences. This pattern weakly (*r* = 0.21, *p* < 0.05) correlated with attachment avoidance (see Appendix A), indicating that individuals with higher attachment avoidance reported more stressful events on average. This suggests a potential bias as avoidant individuals contributed slightly more data compared to their anxious or securely attached counterparts.

Third, in the absence of a validated scale that measures the relative use of interpersonal and intrapersonal ER simultaneously, we introduced a new measure to capture these dimensions. While our measure exhibits face validity as it directly assesses participants’ interpersonal and intrapersonal ER, we recommend developing more validated measures that simultaneously capture interpersonal and intrapersonal ER in both cross-sectional and ESM study designs, similar to a recently developed questionnaire that captured some intrapersonal ER strategies [69].

Fourth, we acknowledge that apart from attachment orientations, other individual, relational, situational, and cultural factors could influence an individual’s use of interpersonal compared to intrapersonal ER strategies. While we controlled for key individual factors (gender, age, and personality traits such as neuroticism and extraversion), as well as relational factors (relationship length and relationship quality) and situational factors (such as availability of others), we recognize that other factors could also have impacted ER behavior. For example, individuals with a more collectivistic cultural background might have used higher levels of interpersonal compared to intrapersonal ER [70]. Additionally, past traumatic relational experiences such as intimate partner violence, emotional abuse, and infidelity might influence the use of interpersonal compared to intrapersonal ER. Such additional influences should be considered in future research. Future investigations could also explore other attachment-related contextual factors such as fatigue, event type (relationship related or not), and the amount of stressfulness of the event [17,40].

Fifth, our study was primarily composed of females, students, and Western participants. Therefore, our findings may not generalize to male populations, individuals with diverse gender identities, and the general population, particularly those of non-Western cultures. Additionally, our sample was predominantly young, in relatively new relationships (with a mean relationship length of 2.5 years in Study 2), unmarried, and without children. These demographic factors could influence attachment-related ER behaviors, meaning our results may not fully apply to older adults, those in longer-term relationships, married individuals, or parents. Future research should aim to include more diverse samples to better understand how these factors interact with attachment orientations and ER. Furthermore, our sample did not include participants with very high levels of attachment insecurity (scores above 5.5 on a scale of 1 = highest security to 7 = highest insecurity). This could limit the generalizability of our findings as in populations with higher levels of attachment insecurity, the dynamics of ER may differ. Future research could explore ER flexibility in populations with very high levels of attachment insecurity.

Notably, future research can further explore alternative ways of measuring variability [71] and alternative conceptualizations and measurements of regulatory flexibility. While our study followed Aldao et al.’s [9] conceptualization of ER flexibility, another significant approach is offered by Bonanno et al. [42], conceptualizing ER flexibility within three inter-related components of context-sensitivity, repertoire, and feedback. Incorporating both frameworks could provide a more comprehensive understanding of ER flexibility [72]. Future research should advance this research by examining the relationship between attachment insecurity and ER flexibility using Bonanno’s conceptualization. Additionally, examining attachment’s association with flexibility in use of specific ER strategies like rumination, suppression, or sharing would be beneficial. 

Additionally, another important factor influencing the use of interpersonal versus intrapersonal ER is the responsiveness of the partner to one’s attempts at interpersonal ER. Future research should advance the knowledge on the effects of partner responsiveness—both enacted and perceived—on interpersonal ER. Dyadic designs involving both partners may be particularly effective in exploring these dynamics. Furthermore, while previous research suggests that in general greater ER flexibility signifies more effective ER [9,40,41], actual investigation into the adaptiveness of higher ER variability and flexibility is recommended [9,73]. Future research could investigate whether the associations we found between attachment insecurities and regulatory behavior translate into adverse ER outcomes, such as heightened experiences of negative emotions. This exploration could elucidate whether the ER difficulties experienced by insecurely attached individuals stem from lower ER flexibility.

## 10. Conclusions

To the best of our knowledge, our study was the first to explore the association between adult attachment and ER flexibility in romantic relationships. To do so, we looked at responses of people in daily life and across different contexts. We found that both attachment anxiety and attachment avoidance are significantly but distinctively associated with different indicators of ER flexibility. Our findings suggest that limited ER flexibility is probably a contributing factor to ER difficulties of individuals with higher attachment insecurity, deepening our understanding of how attachment insecurities affect emotion regulation. For practice, our findings imply that interventions designed to help individuals with attachment insecurity should focus on increasing their ER flexibility. To conclude, our research offers valuable insights into overcoming emotion regulation challenges of insecurely attached people. Ultimately, addressing these difficulties could greatly benefit the well-being of individuals with insecure attachment styles and their close relationships.

## Figures and Tables

**Figure 1 behavsci-14-00758-f001:**
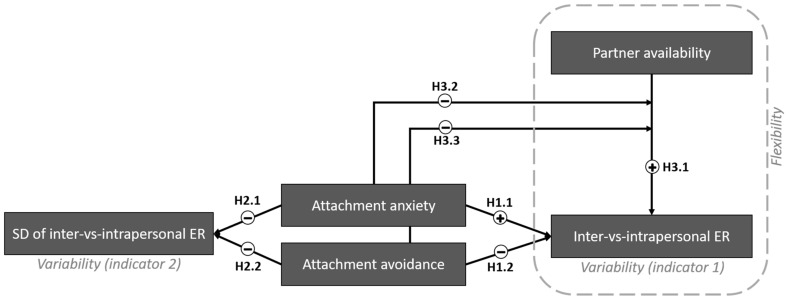
Research hypotheses of Study 1 and Study 2. Note. Study 1 investigated H1.1 and H1.2 in an online cross-sectional survey. Study 2 investigated all hypotheses in an online survey combined with experience sampling method. Inter-vs-intrapersonal ER is interpersonal ER minus intrapersonal ER at a given period of time. SD is standard deviation. The signs in the circles indicate the direction of the hypothesized effect.

**Figure 2 behavsci-14-00758-f002:**
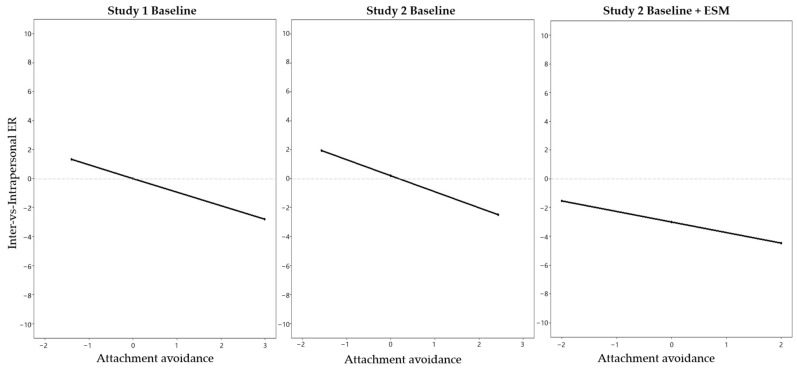
The effect of attachment avoidance on inter-vs-intrapersonal emotion regulation across studies and analyses. Note: the inter-vs-intrapersonal scores closer to 0 indicate more similarity in the use of interpersonal and intrapersonal ER and thus a higher variability. Larger positive or negative scores indicate lower variability.

**Figure 3 behavsci-14-00758-f003:**
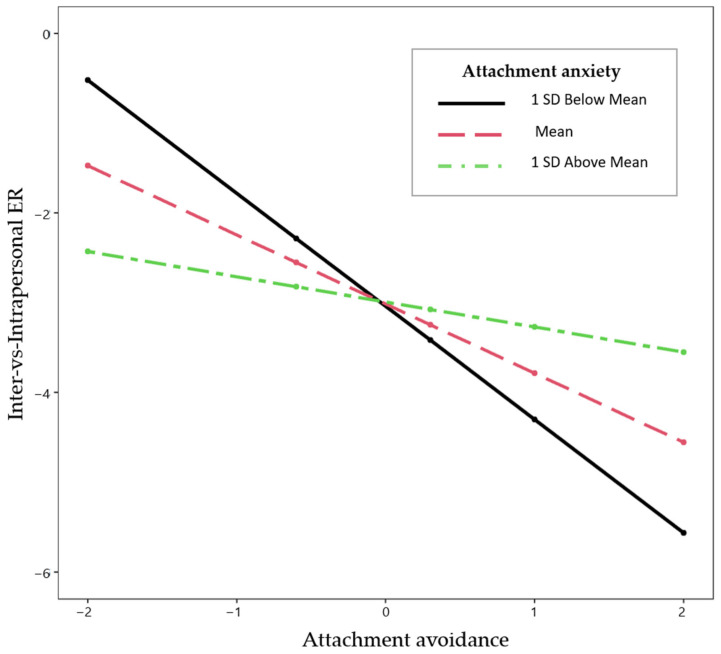
The interaction between attachment anxiety and avoidance affecting inter-vs-intrapersonal emotion regulation.

**Figure 4 behavsci-14-00758-f004:**
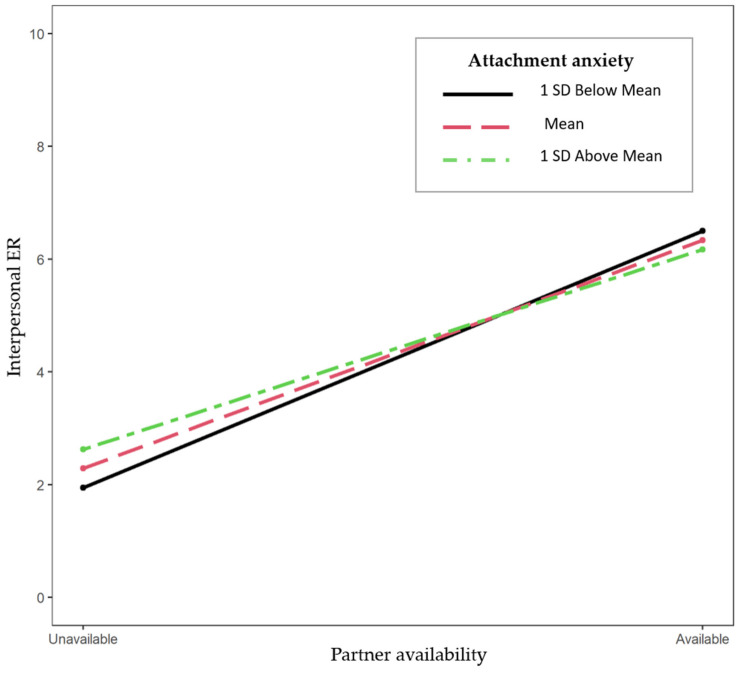
The interaction between attachment anxiety and partner availability affecting interpersonal emotion regulation.

**Table 1 behavsci-14-00758-t001:** Descriptive statistics across studies and analyses.

Variable	Study 1			Study 2 Baseline			Study 2 Baseline + ESM			
	*n*	*M* (*SD*)	Range	*n*	*M* (*SD*)	Range	*n*	*M* (*SD*)	Range	*ICC*
*Baseline measurements*										
Inter-vs-intrapersonal ER	174	−0.01 (1.85)	−4 to 4	287	0.24 (1.90)	−4 to 4	124	0.31 (2.07)	−4 to 4	-
Interpersonal ER	174	3.55 (1.08)	1 to 5	287	3.63 (1.08)	1 to 5	124	3.68 (1.13)	1 to 5	-
Intrapersonal ER	174	3.56 (1.11)	1 to 5	287	3.39 (1.07)	1 to 5	124	3.36 (1.13)	1 to 5	-
Attachment anxiety	174	2.53 (0.88)	1 to 5.06	287	2.60 (1.06)	1 to 6.5	124	2.49 (0.99)	1 to 5.6	-
Attachment avoidance	174	2.40 (0.83)	1 to 5.39	287	2.56 (0.80)	1 to 5	124	2.55 (0.78)	1 to 4.6	-
Age (years)	174	23.79 (7.63)	18 to 58	287	22.13 (5.04)	18 to 58	124	22.45 (6.39)	18 to 58	-
Relationship length (months)	169	42.07 (72.74)	6 to 492	286	30.66 (30.02)	3 to 249	123	33.74 (36.68)	3 to 249	-
Relationship quality	174	5.94 (0.80)	3.17 to 7	282	6.10 (0.79)	2.3 to 7	124	6.18 (0.68)	3.1 to 7	-
Neuroticism	174	4.15 (1.52)	1 to 7	287	4.28 (1.43)	1 to 7	124	4.36 (1.45)	1 to 7	-
Extraversion	174	4.30 (1.73)	1 to 7	287	4.49 (1.47)	1 to 7	124	4.44 (1.51)	1 to 7	-
COVID	174	23.44 (23.67)	0 to 100	287	30.02 (24.49)	0 to 100	124	31.39 (25.18)	0 to 92	-
*Momentary measurements*										
Participant-centered values										
Inter-vs-intrapersonal ER							124	−2.93 (3.05)	−10 to 5.4	-
Interpersonal ER							124	3.58 (1.93)	0 to 9	-
Intrapersonal ER							124	6.51 (1.66)	0.7 to 10	-
Stressfulness of event							124	5.12 (1.56)	1.6 to 8.4	-
Partner availability							124	0.32 (0.23)	0 to 0.9	-
Others availability							124	0.42 (0.21)	0 to 0.9	-
Grand values										
Inter-vs-intrapersonal ER							2654	−3.1 (5.62)	−10 to 10	0.22
Interpersonal ER							2663	3.5 (3.66)	0 to 10	0.21
Intrapersonal ER							2654	6.61 (2.96)	0 to 10	0.25
Stressfulness of event							2666	5.12 (2.52)	0 to 10	0.35
Partner availability							2672	0.31 (0.46)	0 to 1	0.20
Others availability							2672	0.41 (0.49)	0 to 1	0.14

**Table 2 behavsci-14-00758-t002:** The effect of attachment on the use of interpersonal compared to intrapersonal emotion regulation across studies and analyses.

	Study 1 Baseline (*N* = 174)				Study 2 Baseline (*N* = 287)				Study 2 ESM (*N* = 124)				
	*b*	*SE*	*t*	*p*	*b*	*SE*	*t*	*p*	*b*	*SE*	*DF*	*t*	*p*
(Intercept)	**−0.69**	**0.28**	**−2.45**	**0.016**	−0.41	0.28	−1.48	0.139	**−5.67**	**0.55**	**2483**	**−10.38**	**<0.001**
Attachment anxiety	−0.13	0.18	−0.73	0.464	0.02	0.12	0.14	0.885	0.03	0.26	113	0.10	0.922
Attachment avoidance	**−0.94**	**0.21**	**−4.37**	**<0.001**	**−1.11**	**0.14**	**−7.78**	**<0.001**	**−0.77**	**0.32**	**113**	**−2.45**	**0.016**
Attachment anxiety × Attachment avoidance	−0.02	0.17	−0.13	0.900	0.15	0.11	1.395	0.164	**0.50**	**0.23**	**113**	**2.13**	**0.035**
Relationship length	−0.01	0.01	−0.34	0.732	0.00	0.00	0.17	0.869	−0.01	0.01	113	−1.57	0.120
Female	**0.92**	**0.33**	**2.81**	**0.006**	**0.74**	**0.30**	**2.46**	**0.015**	0.41	0.57	113	0.72	0.471
Others					−0.86	1.02	−0.85	0.399					
Relationship quality	−0.38	0.23	−1.71	0.090	−0.27	0.15	−1.71	0.088	0.05	0.36	113	0.13	0.899
Neuroticism	**−0.23**	**0.09**	**−2.47**	**0.015**	−0.14	0.08	−1.73	0.084	0.28	0.16	113	1.76	0.081
Extraversion	−0.03	0.08	−0.44	0.661	**0.15**	**0.07**	**2.13**	**0.034**	0.13	0.14	113	0.96	0.339
COVID	−0.01	0.01	−0.79	0.432	−0.01	0.00	−1.38	0.169	0.02	0.01	113	1.83	0.070
Stressfulness of event									**0.15**	**0.06**	**2483**	**2.45**	**0.014**
Partner availability									**5.89**	**0.36**	**2483**	**16.48**	**<0.001**
Others availability									**1.10**	**0.24**	**2483**	**4.56**	**<0.001**
Time									−0.01	0.01	2483	−1.39	0.166

Note: the significant results are presented in bold.

**Table 3 behavsci-14-00758-t003:** Analysis of attachment insecurities affecting variability in the use of inter-vs-intrapersonal emotion regulation across momentary assessments.

Independent Variable	*b*	*SE*	*t*	*p*
(Intercept)	**4.41**	**0.62**	**7.70**	**<0.001**
Attachment anxiety	**−0.46**	**0.18**	**−2.56**	**0.012**
Attachment avoidance	0.09	0.23	0.39	0.699
Attachment anxiety × Attachment avoidance	0.25	0.17	1.50	0.137
Relationship length	−0.01	0.00	−1.48	0.143
Relationship quality	0.12	0.27	0.44	0.660
Neuroticism	−0.11	0.12	−0.97	0.336
COVID	0.01	0.01	0.94	0.351
Participant level mean stressfulness of events	−0.08	0.10	−0.85	0.397
Participant level mean partner availability	**1.62**	**0.70**	**2.36**	**0.020**
Participant level mean others availability	0.50	0.72	0.69	0.490

Note. The significant results are presented in bold.

**Table 4 behavsci-14-00758-t004:** Moderation effect of attachment on the association between partner availability and inter-vs-intrapersonal emotion regulation.

Independent Variable	Inter-vs-Intrapersonal ER				
	*b*	*SE*	*DF*	*t*	*p*
(Intercept)	**−5.70**	**0.55**	**2480**	**−10.44**	**<0.001**
Attachment anxiety	0.19	0.28	113	0.67	0.505
Attachment avoidance	**−0.87**	**0.35**	**113**	**−2.51**	**0.013**
Partner availability	**5.98**	**0.36**	**2480**	**16.43**	**<0.001**
Attachment anxiety: Attachment avoidance	**0.58**	**0.26**	**113**	**2.26**	**0.026**
Attachment anxiety: Partner availability	−0.55	0.37	2480	−1.50	0.133
Attachment avoidance: Partner availability	0.29	0.46	2480	0.63	0.527
Attachment anxiety: Attachment avoidance: Partner availability	−0.38	0.41	2480	−0.92	0.357
Relationship length	−0.01	0.01	113	−1.57	0.119
Female	0.42	0.57	113	0.74	0.462
Relationship quality	0.04	0.36	113	0.12	0.904
Neuroticism	0.28	0.16	113	1.73	0.087
Extraversion	0.14	0.14	113	1.00	0.320
COVID	0.02	0.01	113	1.87	0.065
Stressfulness of event	**0.15**	**0.06**	**2480**	**2.44**	**0.015**
Others availability	**1.11**	**0.24**	**2480**	**4.58**	**<0.001**
Time	−0.01	0.01	2480	−1.41	0.159

Note: the significant results are presented in bold.

## Data Availability

The data presented in this study as well as analytical codes will be openly available in the Open Science Framework (https://osf.io/et5a2) (accessed on 18 August 2024).

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
