# Peer review of "Adult Attachment and Emotion Regulation Flexibility in Romantic Relationships"

_behavsci, 2024, doi:10.3390/bs14090758_

Round 1

Reviewer 1 Report

Comments and Suggestions for Authors

This paper examines the relationship between adult attachment and emotion regulation (ER) flexibility in a sample of individuals in monogamous romantic relationships.

The results presented in this paper are of potential importance in elucidating the complex relationship between childhood attachment, adult attachment style, emotion regulation, and a wide range of mental health, relational and social outcomes. The arguments and hypotheses presented by the researchers are cogent, and the study methodology is sound overall. The authors have also acknowledged the limitations of their work in terms of representativeness of the sample (female preponderance, Western culture).

The following are aspects of the paper that could benefit from correction or clarification:

1. Interpersonal ER can also draw on persons other than a romantic partner, if they are available (parents, siblings, friends, adult children, even therapists or counselors.) Was any attempt made to correct for the confounding effect of this variable?

2. Apart from attachment style, there are other factors that could influence an individual's likelihood of utilizing intrapersonal vs. interpersonal ER strategies, including individual temperament (harm avoidance, reward dependence) and cultural / social background. How might these factors have affected the study's findings?

3. The authors have mentioned the high number of women in the study sample as a limitation of the study. Were sub-group analyses considered to examine whether men and women differed in terms of ER strategies used? What does the existing literature have to contribute in this regard? In the data analyses, it was mentioned that gender was entered into the models - was there any independent effect of gender on ER flexibility?

4. This study has used a method derived from previous research (Aldao et al. 2015) to assess ER flexibility. Has this method been validated by independent researchers other than those involved in the current study? Are there any other methods of assessing ER flexibility? What are the advantages and disadvantages of the method chosen for this research?

5. There is a very wide range in relationship length, which could independently influence a person's likelihood to rely on their partner for interpersonal ER. Was any independent effect of relationship length on choice of ER strategy? Did prior relationship status (i.e., participant in romantic relationships / marriages prior to the current one) affect the likelihood of using interpersonal ER? What about past trauma (intimate partner violence, emotional abuse, infidelity) in the context of a relationship?

6. The partner's responsiveness to the participant's needs could also influence their tendency to rely on them for interpersonal ER. This, in turn, could depend on the partner's attachment style (e.g., a partner with high attachment avoidance or anxiety may be reluctant to "provide" interpersonal support beyond a certain point.) Could this have been assessed in any way, even if the partners were excluded from participation in the main study?

7. Did any of the participants have a physical or mental illness or disability that could influence their choice of ER strategy? (If this was an exclusion criterion for the participants, this should be clearly stated in the Methods section.)

8. The current study has used a relatively "well" sample with rather low levels of attachment insecurity. Could this have led to a "null" result, or an underestimation of the effects hypothesized by the authors? 

9. What are the implications of the current research for practice and research? This could be developed a little further in the Discussion and Conclusions sections.

Reviewer 2 Report

Comments and Suggestions for Authors

Thank you for the opportunity to review this interesting and well-written manuscript which reports two studies that used survey methods (Study 1) and experience sampling methods (Study 2) to investigate how adult attachment orientation impacts the use of intrapersonal and interpersonal emotion regulation strategies. The authors found that higher attachment avoidance is associated with the tendency to rely on intrapersonal emotion regulation strategies (over interpersonal emotion regulation strategies), whereas higher attachment anxiety showed more nuanced links with emotion regulation flexibility.

The introduction and discussion are comprehensive and well-written. The discussion does a good job of presenting each result within the context of prior literature. The methods and results of each study are also reported clearly. The authors provide a reasonable justification for calculating their inter-vs-intrapersonal ER measure. Use of covariates is also well-justified.

In Section 11.5, the authors should also mention the limitation of their samples being predominantly young, in newer relationships (mean of 2.5 years in Study 2), unmarried, and without children as each of these factors could influence the generalizability of their results.
